# Durum Wheat Yield and N Uptake as Affected by N Source, Timing, and Rate in Two Mediterranean Environments [†]

Silvia Pampana [1,*] and Marco Mariotti [2]

1   Department of Agriculture, Food and Environment, University of Pisa, 56124 Pisa, Italy
2   Department of Veterinary Science, University of Pisa, 56124 Pisa, Italy; marco.mariotti@unipi.it
*   Correspondence: silvia.pampana@unipi.it; Tel.: +39-050-2218941
†   Invited paper at the 1st International Electronic Conference on Agronomy, 3–17 May 2021; Available online: https://sciforum.net/conference/IECAG2021 (accessed on 18 June 2021).

**Abstract:** In nitrate vulnerable zones (NVZs), site-specific techniques are needed to match N availability with durum wheat (*Triticum turgidum* subsp. durum Desf.) requirements. Enhanced-efficiency fertilizers can improve efficient N supply and reduce leaching, contributing to sustainable agriculture. Two-year field experiments were carried out at two Mediterranean nitrate vulnerable zones in Central Italy (Pisa and Arezzo) to study the effects of nitrogen sources, timings, and application rates. The trial compared: (i) three N sources for the first topdressing application (urea, methylene urea, and urea with the nitrification inhibitor DMPP); (ii) two stages for the first topdressing N application (1st tiller visible—BBCH21 and 1st node detectable—BBCH31); (iii) two N rates: one based on the crop N requirements (Optimal—$N_O$), the other based on action programme prescriptions of the two NVZs (Action Programme—$N_{AP}$). Grain yield and yield components were determined, together with N uptake. The results showed that: (i) grain and biomass production were reduced with $N_{AP}$ at both locations; (ii) urea performed better than slow-release fertilizers; (iii) the best application time depended on the N source and location: in Pisa, enhanced-efficiency fertilizers achieved higher yields when applied earliest, while for urea the opposite was true; in Arezzo different N fertilizers showed similar performances between the two application timings. Different behaviors of topdressing fertilizers at the two localities could be related to the diverse patterns of temperatures and rainfall. Thus, optimal fertilization strategies would seem to vary according to environmental conditions.

**Keywords:** 3,4-dimethylpyrazole phosphate; durum wheat; environmental impact; methylene urea; nitrogen management; nitrate-vulnerable zones; sustainable agriculture; urea

## 1. Introduction

Nitrogen (N) is a major macronutrient that often limits plant growth. Therefore, crop yield and quality depend greatly on extensive inputs of fertilizer nitrogen for sustainable and profitable crop production. However, N fertilization has environmental impacts associated with nitrate leaching, eutrophication, and global warming, due to the emission of nitrous oxide gases [1].

To prevent and reduce water pollution by nitrates from agricultural sources, the European Union (EU) introduced the Nitrate Directive (ND) (91/676/EEC). They are a set of actions, defined at the regional level, obliging the member states to designate areas vulnerable to nitrate pollution (nitrate vulnerable zones—NVZs). In NVZs, farmers must follow a range of measures, including controlling the timing and quantities of fertilizers applied to the land [2].

In the Mediterranean areas, year-to-year rainfall variability makes it difficult to establish definite agronomic practices. The potential for leaching of fall-applied nitrogen is high due to heavy rainfall associated with still limited crop needs [3]. Thus, aquifers' non-point source nitrate pollution is regarded as one of the main agricultural impacts, and N leaching is the major determinant of the low N use efficiency (NUE) of crops.

Durum wheat (*Triticum turgidum* L. subsp. durum) is the most cultivated winter crop in the Mediterranean basin. It is typically sown in late autumn or early winter and harvested in late spring-early summer. Mediterranean soils tend to be poor in organic matter and total nitrogen content; therefore, the crop requires the intensive use of N fertilizers to achieve sufficient yields and good grain quality [4]. In these areas, due to high rainfall and low crop evapotranspiration rates in autumn, low N use efficiencies of N fertilizers have been reported, increasing the risks of N leaching losses [5].

NUE is considered the efficiency of nitrogen recovery from the applied fertilizer, or the N available to the crop [6], or as a productivity index, expressed as the yield produced per unit of available N [7,8]. Whichever definition is used for NUE determination, it relates production as a function of inputs; thus, given a constant input, any yield increase will be reflected in a greater NUE. Accordingly, N uptake is a second-level trait influencing N efficiency [9]. In addition, the nitrogen harvest index (NHI) and the N content are fairly important nitrogen indexes for evaluating fertilizers in crops, such as durum wheat, for which N absorption (and therefore the resulting protein content) needs to be calculated [10].

In Mediterranean nitrate-vulnerable zones, a fertilizer application that matches N supply with crop demand is even more imperative for the efficient use of N. Nitrogen fertilization should be fine-tuned, and enhance both yield and quality and reduce environmental risks [11], by combining the: (i) source of N application; (ii) timing; and (iii) rate.

Enhanced-efficiency fertilizers (EEFs) could be useful for synchronizing N release from fertilizers with N uptake by crops, enhancing N use efficiency and reducing losses to the environment [12].

Of these, slow-release fertilizers (SRFs) are long-chain molecules with little solubility, such as formaldehyde, isobutylene diurea, or methylene urea (MU). MU is a condensation product of urea and formaldehyde consisting of polymers with various chain lengths, which promote the slow-release of N [13].

Other EEFs are stabilized nitrogen fertilizers that contain nitrification inhibitors (NI) which slow the rate at which urea is hydrolyzed in the soil [14]. One of the most commonly used nitrification inhibitors is 3,4-dimethylpyrazole phosphate (DMPP). It delays ammonium ($NH_4^+$) conversion to nitrate ($NO_3^-$), blocking ammonia monooxygenase, the enzyme that catalyzes the first rate-limiting step of nitrification.

Field studies have shown that the efficiency of these fertilizers can vary significantly depending on the environmental conditions because soil water content and temperature are responsible for variation in the efficiency of nitrification inhibitors [14,15].

In central Italy, the recommended timing for the first topdressing N application to durum wheat is between late tillering and the onset of stem elongation. This is because the differentiation of the first initiated spikelet's starts before the first node becomes detectable. When elongation of culm internodes becomes visible, spike size, in terms of the number of spikelets, is already defined [3]. Due to climate change, heavier and more frequent rainfall can produce excessive soil moisture, and thus, fertilization should be postponed. Delaying N fertilizer application may have adverse effects on crop yield, hampering some yield determinants during the early growth stages, such as the production of leaf area and the number of grains per unit area.

Crop N use efficiency in durum wheat has been demonstrated to be low also because the N fertilizer rate often exceeds crop needs [16]. However, under proper soil N levels, reduced N rates and split N applications between fall and spring can maintain high durum wheat yields [3,5]. Moreover, for this reason, in NVZs, the application of inorganic N fertilizers is limited by crop type at lower than optimal N rates. These rates are calculated based on a balanced approach to cover the crop demand for a target grain yield and correlated to environmental conditions.

Split applications of N fertilizers also improve N use efficiency. However, the crop response is conditioned by climate and agronomic practices, such as quantity, splitting ratios, and timings of fertilizer applications together with the type of fertilizer used [3,17].

Since durum wheat productivity and N fertilizer use can strongly differ among locations due to the variability in pedoclimatic factors, the effects of different N management approaches should be site-specifically evaluated in each NVZ to optimize N fertilization.

We hypothesized that the application of slow-release fertilizer (MU) or a fertilizer with a nitrification inhibitor (NI) could precede the first topdressing N fertilization at the tillering of durum wheat, preventing yield drawbacks and losses of unused N.

Overall, we evaluated: (i) the effects on grain yield and N uptake of the topdressing N application of three N fertilizers to durum wheat at two different growth stages and at two N rates. (ii) Whether the different environments of two NVZs potentially influenced the stage at which the first N application could be applied.

## 2. Materials and Methods

The research was carried out in open fields from November 2010 to June 2011 (2010 hereafter) and from November 2011 to June 2012 (2011 hereafter) at two experimental stations located in two different NVZs in Tuscany, central Italy. They included (i) the Research Centre of the Department of Agriculture, Food and Environment of the University of Pisa, (43°40 N, 10°19 E) (Pisa); and (ii) the Research Centre for Agricultural Technologies at Cesa, Arezzo (43°18 N, 11°48 E) (Arezzo). The climate of both sites is hot-summer Mediterranean (Csa).

In Pisa, the long-term mean annual maximum and minimum daily air temperatures are 20.2 °C and 9.5 °C, and the mean rainfall is 971 mm year$^{-1}$, with 515 mm received during durum wheat cultivation (November–July). The characteristics of the first 30 cm of the loam soil at Pisa were: 44.6% sand (2 mm > Ø > 0.05 mm), 41.1% silt (0.05 mm > Ø > 0.002 mm), 14.3% clay (Ø < 0.002 mm); 8.1 pH; 2.0% organic matter (Walkley and Black method); 1.1 g kg$^{-1}$ total nitrogen (Kjeldahl method); 9.9 mg kg$^{-1}$ available P (Olsen method); 145.3 mg kg$^{-1}$ available K (BaCl$_2$ + TEA method).

In Arezzo, the annual maximum and minimum daily air temperatures are 19.8 °C and 8.7 °C, respectively, and the total annual rainfall is 755 mm, with 499 mm during the wheat growing cycle. The physical-chemical properties of the first 30 cm of the silty clay loam soil were: 17.7% sand (2 mm > Ø > 0.05 mm); 49.8% silt (0.05 m > Ø > 0.002 mm); 32.5% clay (Ø < 0.002 mm); 7.7 pH; 1.3% organic matter (Walkley and Black method); 2.7 g kg$^{-1}$ total nitrogen (Kjeldahl method); 25.0 mg kg$^{-1}$ available P (Olsen method); 155 mg kg$^{-1}$ available K (BaCl$_2$ + TEA method).

In both years, daily weather data were obtained from meteorological stations located in the experimental fields. Throughout the experiment, phenological phases were recorded using the BBCH scale for cereals [18] to determine N application periods and harvesting times (Table 1).

**Table 1.** Durum wheat growth stages in the two growing seasons (2010 and 2011) at the two locations.

| Stage | BCCH | Pisa First Season | Pisa Second Season | Arezzo First Season | Arezzo Second Season |
|---|---|---|---|---|---|
| Sowing | 00 | 25 November 2009 | 28 November 2010 | 25 November 2009 | 28 November 2010 |
| Tillering [1] | 21 | 10 February 2010 | 19 February 2011 | 6 March 2010 | 10 March 2011 |
| 1st node [1] | 31 | 25 March 2010 | 28 March 2011 | 8 April 2010 | 8 April 2011 |
| 2nd node [2] | 32 | 9 April 2010 | 13 April 2011 | 22 April 2010 | 26 April 2011 |
| Full Flowering | 65 | 2 May 2010 | 6 May 2011 | 12 May 2010 | 16 May 2011 |
| Maturity | 99 | 12 July 2010 | 16 July 2011 | 21 July 2010 | 23 July 2011 |

[1] 1st topdressing application; [2] 2nd topdressing application.

The crop was grown following a standard technique for central Italy except for N fertilization. The soil was plowed at 40 cm depth in September; final seedbed preparation was carried out immediately before sowing by harrowing twice, with a disc harrow and with a rotating harrow.

Sowing of the variety Latinur of durum wheat was performed at both locations using a plot drill at the rate of 400 seeds m$^{-2}$ on 25 and 28 November 2010 and 2011, within the

optimum sowing time for wheat production in central Italy (Table 1). Phosphorus and potassium were applied before seeding as triple mineral phosphate and potassium sulfate at 100 kg ha$^{-1}$ $P_2O_5$ and 100 kg ha$^{-1}$ $K_2O$. Weed control was performed at the stage of 4th–5th leaf un-folded by distributing commercial herbicides.

The trial compared: (i) three N sources for the first topdressing application (urea, methylene urea (MU), and urea with the nitrification inhibitor (NI) DMPP); (ii) two stages for the first topdressing N application (1st tiller visible—BBCH21 and 1st node detectable—BBCH31); (iii) two N rates: one based on the crop N requirements (Optimal—$N_O$), the other based on the Action Programme prescriptions of the two NVZs (Action Programme—$N_{AP}$).

A randomized complete block design was used for each year and location, with treatments in a split-split-plot arrangement with three replicates. N sources for the first topdressing application were the main plots, times for the first topdressing application were allocated as sub-plots and N rates as sub-sub-plots. Each year, 36 plots were prepared at both locations, comparing 12 treatments replicated three times (3 N fertilizers × 2 N application times × 2 N rates × 3 replications).

The optimal rates ($N_O$) were calculated following the balance method to achieve target yields of 5 and 6 Mg ha$^{-1}$ in Pisa and Arezzo, respectively, resulting from the long-term averages of the two areas, both with 13.5% of protein. Accordingly, at the two locations, the optimal rates were 160 and 190 kg N ha$^{-1}$, respectively [3,4]. Correspondingly, the N rates based on action programme prescriptions ($N_{AP}$) were 100 and 112 kg N ha$^{-1}$ at the two NVZs. Therefore, total N rates ($N_O$ and $N_{AP}$) were split into three applications: 30 kg N ha$^{-1}$ at sowing and the remaining split into two equal topdressing applications: the first at tillering (BBCH21) or at the 1st node detectable (BBCH31), and the second at the 2nd node detectable (BBCH32).

The fertilizers applied were: (i) ammonium sulfate at sowing; (ii) urea, methylene urea (MU), and urea with a nitrification inhibitor (NI) 3,4-dimethyl pyrazole phosphate—DMPP at the first topdressing application (BBCH21 or BBCH31); (iii) urea at the second topdressing application (BBCH32).

At physiological maturity (BCCH99), plants from four adjacent rows of 1 m length were manually cut at the ground level and partitioned into culms, leaves, and spikes. Dry weights of vegetative above-ground plant parts (VAP) were determined, and spikes were counted and subsequently separated into kernels and chaff. Mean kernel weight (MKW) and the number of kernels per unit area were also determined, and harvest index (HI) was calculated as the ratio of grain yield to total above-ground biomass.

The dry weight (DW) of all plant parts was measured by oven-drying at 65 °C to a constant weight. All plant parts were analyzed for N concentration using the micro-Kjeldahl standard method. Total nitrogen uptake was obtained by multiplying N concentrations of different plant parts by DW.

The nitrogen harvest index was obtained as the ratio of N content in grains to the above-ground N content.

Data were initially checked to verify the normality and homogeneity of variance assumptions, then ANOVA over the two years was carried out for each location. The main effects of the year (Y), type of N fertilizer (S) at the first topdressing application, time of the first topdressing application (T), N rate (R), and their interactions were tested for dry weight of plant parts and relative N concentration and content. Significantly different means were separated at the 0.05 probability level by the least-significant difference test [19].

## 3. Results

### 3.1. Weather Conditions

At the two locations, temperatures were similar in both years and close to the long-term average (Figure 1). Maximum and minimum temperatures were higher at Pisa than at Arezzo in winter; in spring, maximum temperatures were similar at the two sites, while minimum temperatures were lower at Arezzo, with the maximum difference being in the last ten days of March in both years.

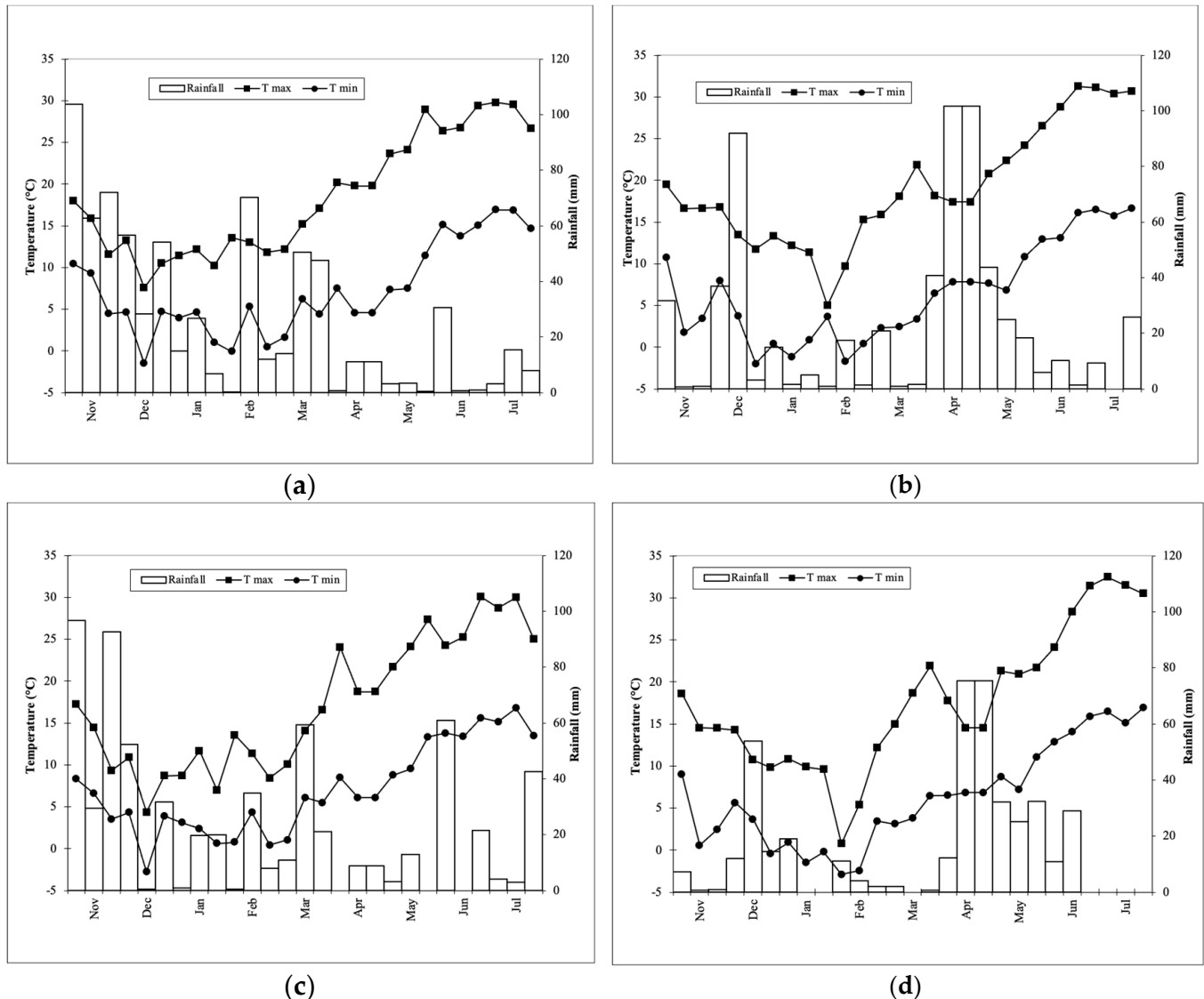

**Figure 1.** Maximum and minimum temperature and rainfall recorded in the two durum wheat growing seasons at the two locations: (**a**) Pisa first season (November 2010–June 2011); (**b**) Pisa second season (November 2011–June 2012); (**c**) Arezzo first season (November 2010–June 2011); (**d**) Arezzo second season (November 2011–June 2012).

At Pisa, rainfall during the crop cycle was similar in both years (about 475 mm) and slightly lower than the long-term average (515 mm). At Arezzo, rainfall differed between years and was 413 mm in 2010 and 351 mm in 2011, correspondingly similar to 30% lower than the long-term average (499 mm).

*3.2. Grain Yield*

For some of the parameters analyzed, the analysis of variance revealed significant differences among treatments at the two sites (Table 2). Below we report and discuss the main results.

**Table 2.** Results of ANOVA for durum wheat vegetative above-ground part (VAP), grain yield, yield components, and NHI and N uptake as affected by Year (Y), N source (S), N timing (T), N rate (R), and their interactions at the two locations. * = significant at 0.05 level; ns = non-significant.

| | VAP | Grain | H.I. | Spikes | Kernels | MKW | NHI | N Uptake |
|---|---|---|---|---|---|---|---|---|
| | Mg ha$^{-1}$ | Mg ha$^{-1}$ | % | n m$^{-2}$ | n spike$^{-1}$ | mg | % | kg ha$^{-1}$ |
| **Pisa** | | | | | | | | |
| Y | * | * | ns | * | * | * | ns | * |
| S | * | * | ns | * | * | ns | ns | ns |
| T | * | ns | * | ns | ns | ns | ns | ns |
| R | * | * | ns | ns | * | * | * | * |
| Y × S | ns | ns | ns | ns | ns | ns | ns | ns |
| Y × R | ns | ns | ns | ns | ns | ns | ns | ns |
| Y × T | ns | ns | ns | ns | ns | ns | ns | ns |
| S × R | ns | * | ns | * | ns | ns | ns | * |
| S × T | * | * | ns | * | ns | ns | ns | ns |
| R × T | ns | ns | ns | ns | ns | ns | ns | ns |
| Y × S × R | ns | ns | ns | ns | ns | ns | ns | ns |
| Y × S × T | ns | ns | ns | ns | ns | ns | ns | ns |
| Y × R × T | ns | ns | ns | ns | ns | ns | ns | ns |
| S × R × T | ns | ns | ns | * | * | * | ns | ns |
| Y × S × R × T | ns | ns | ns | ns | ns | ns | ns | ns |
| **Arezzo** | | | | | | | | |
| Y | * | * | ns | * | ns | * | ns | ns |
| S | ns | ns | ns | ns | ns | ns | ns | ns |
| T | ns | ns | ns | ns | ns | ns | ns | ns |
| R | * | * | ns | * | ns | * | ns | * |
| Y × S | ns | ns | ns | ns | ns | ns | ns | ns |
| Y × R | ns | ns | ns | ns | ns | ns | ns | ns |
| Y × T | ns | ns | ns | ns | ns | ns | ns | ns |
| S × R | ns | ns | ns | ns | ns | * | ns | * |
| S × T | ns | ns | ns | ns | ns | ns | ns | ns |
| R × T | ns | * | ns | ns | * | ns | ns | ns |
| Y × S × R | ns | ns | ns | ns | ns | ns | ns | ns |
| Y × S × T | ns | ns | ns | ns | ns | ns | ns | ns |
| Y × R × T | ns | ns | ns | ns | ns | ns | ns | ns |
| S × R × T | ns | ns | ns | ns | ns | ns | ns | ns |
| Y × S × R × T | ns | ns | ns | ns | ns | ns | ns | ns |

### 3.2.1. Year Effect

There were significant differences between years at both locations for some of the measured parameters, but none of the interactions with the year was significant (Table 2).

Durum wheat was affected similarly by year at the two NVZs. The crop produced 30% more dry biomass in vegetative above-ground parts (VAP) and 39% higher grain yields in the first season at Pisa, 24% and 27% at Arezzo, respectively (Table 3). At both sites, yield rise was due to increased spikes produced per unit area (2-fold at Pisa and about +14% at Arezzo) and to slight increases (less than 10%) in the MKW, which together compensated for the lower number of kernels per spike at Pisa (−37%). At Arezzo, kernels per spike were the same in both years.

**Table 3.** Dry weight of vegetative above ground parts (VAP), grain yield, Harvest Index (HI), yield components and Nitrogen Harvest Index (NHI) and total N uptake as affected by Year (Y), at Pisa and Arezzo. For each location, values followed by different letters within lines are significantly different, values followed by ns are not significantly different ($p < 0.05$).

| Character | u.m. | Pisa | | | | Arezzo | | | |
|---|---|---|---|---|---|---|---|---|---|
| | | **2010** | | **2011** | | **2010** | | **2011** | |
| VAP | Mg ha$^{-1}$ | 4.3 | a | 3.3 | b | 6.5 | a | 5.2 | b |
| Grain | Mg ha$^{-1}$ | 3.5 | a | 2.5 | b | 5.2 | a | 4.1 | b |
| HI | % | 44.7 | a | 43.1 | b | 44.6 | a | 43.9 | b |
| Spikes | n m$^{-2}$ | 405.6 | a | 194.3 | b | 461.8 | a | 338.0 | b |
| Kernels | n spike$^{-1}$ | 24.1 | b | 38.1 | a | 29.2 | b | 33.8 | a |
| MKW | mg | 35.5 | a | 33.8 | b | 38.5 | a | 35.6 | b |
| NHI | % | 73.7 | ns | 72.1 | ns | 73.6 | ns | 72.8 | ns |
| N uptake | kg ha$^{-1}$ | 83.5 | a | 70.8 | b | 124.8 | a | 114.2 | b |

The crop's total N uptake was similarly boosted in the first season (+18 and 9% respectively at Pisa and Arezzo), even if NHI did not differ between years at the two locations (Table 3).

### 3.2.2. N Source, N Rate and N Timing Effects

The type of fertilizer used (N source) at the first topdressing N application led to differences in the vegetative above-ground part and grain yield in Pisa but not in Arezzo (Table 2). In Pisa, the N source produced different effects depending on the rate and timing of application. N source x N rate and N source × N timing interactions were significant in determining durum wheat production (Table 2). Grain yield was higher with the optimal rate only when fertilization was performed with urea; it did not change when MU or NI was used (Figure 2a).

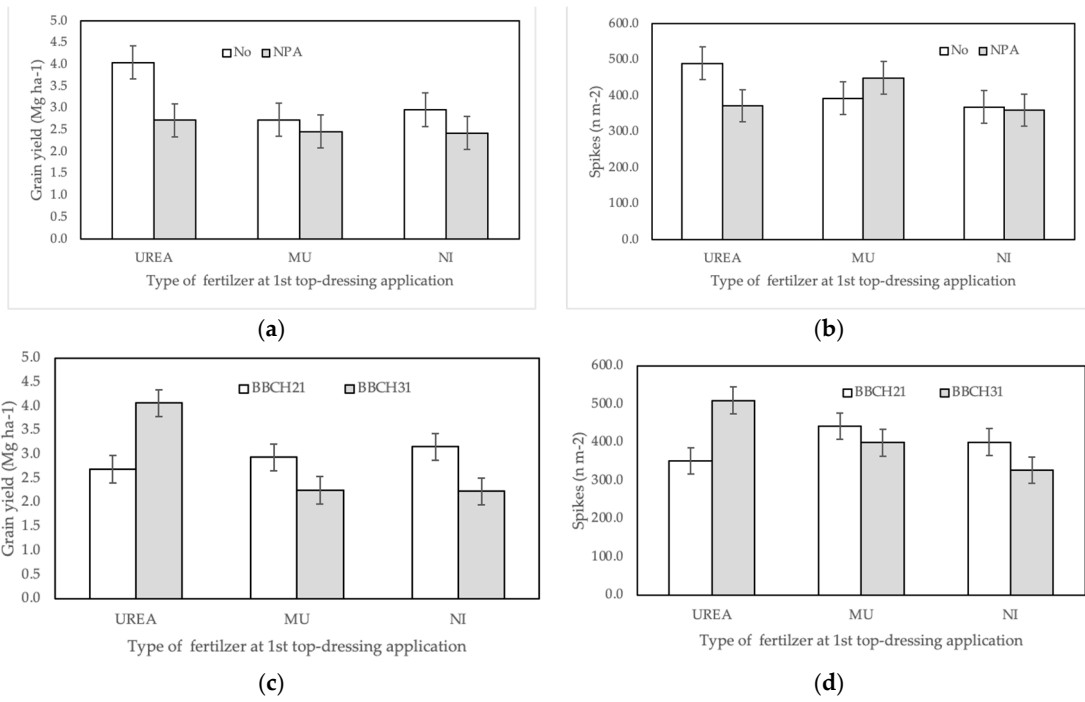

**Figure 2.** N source interaction effects at Pisa: (**a**) Grain yield (N source × N rate interaction); (**b**) Spike number per unit area (N source × N rate interaction); (**c**) Grain yield (N source × N timing interaction); (**d**) Spike number per unit area (N source × N timing interaction). Vertical bars represent LSD ($p = 0.05$).

The same was true for the timing of the first topdressing fertilization (Figure 2c): urea performed better when applied at 1st node detectable (BBCH31), while MU and NI fertilizers produced higher yields at the earlier stage. Both effects were due to the more spikes developed by the crop, which were most abundant when urea was applied at $N_O$ rate (Figure 2b) and BCCH31 (Figure 2d).

In Arezzo, the N source did not significantly influence any of the parameters (Table 2), neither the N source × N timing interaction was significant. Conversely, at this location, grain yield was affected by N rate × N timing interaction (Table 2): the N optimal rate prompted a 12% higher grain yield when the 1st topdressing application was at BCCH21, mainly due to 21% more kernels per spike (Figure 3a,b).

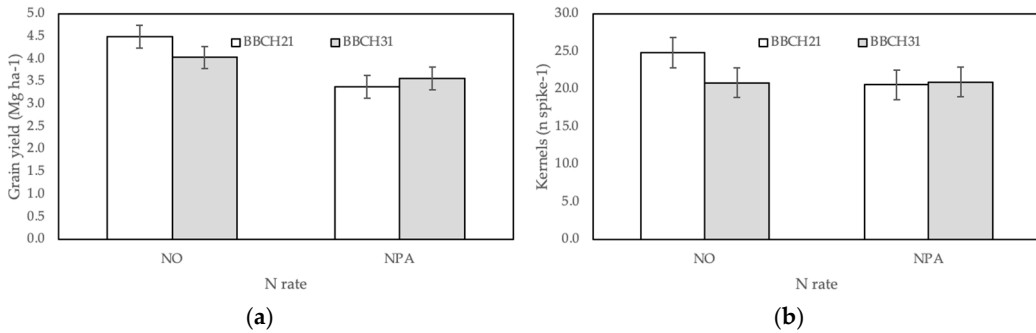

**Figure 3.** N rate × N timing interaction effect at Arezzo: (**a**) Grain yield; (**b**) Kernel number per spike. Vertical bars represent LSD (*p* = 0.05).

No differences were detected between the two growth stages of N application with the action programme rate ($N_{AP}$).

As predicted, higher N rates increased grain yield at both sites (Table 2), and, averaged over the years, source, and timings, optimal N rates ($N_O$) produced grain yields of 3.9 Mg ha$^{-1}$ at Pisa and 5.7 Mg ha$^{-1}$ at Arezzo. This corresponded to rises of about 25% (+28 and +23%) compared to the $N_{AP}$ rates, which yielded 3.1 and 4.7 Mg ha$^{-1}$ respectively.

The higher yield with the $N_O$ was due to more kernels produced per spike (27.0 with $N_O$ and 21.2 with $N_{AP}$), which compensated for a slightly lower MKW (34.6 mg instead of 36.5 mg) at the Pisa site. Meanwhile, at the Arezzo site, the lower MKW (37.4 mg versus 39.6 mg) was compensated for by more spikes produced per unit area (500.2 spikes m$^{-2}$ with $N_O$ and 423.3 with $N_{AP}$).

### 3.3. Nitrogen Harvest Index and Nitrogen uptake

At the Pisa site, the partitioning of nitrogen between grain and straw was changed only by the amount of N given (N rate) (Table 2). The optimal N rate caused a lower NHI (69.9%) than the Action Programmes rate (74.2%), while at the Arezzo site, NHI was not affected by any of the treatments.

Total N uptake of the crop changed between years in Pisa, increasing by 18% in the first season (83.5 kg ha$^{-1}$ in 2010 vs. 70.8 kg ha$^{-1}$ in 2011) due to the higher biomass produced. However, N uptake was not significantly different in the two experimental seasons in Arezzo (Table 2).

What is more, the N source differently affected the N uptake of the crop at both locations, depending on the total N applied (N source × N rate interaction—Table 2).

At both sites, durum wheat maximized the N uptake when urea was applied at the 1st topdressing event and the optimal N rate ($N_O$). This treatment increased the N uptake of durum wheat by 54% and 31% at Pisa and Arezzo, respectively, compared to the reduced N rate ($N_{AP}$) (Figure 4a,b).

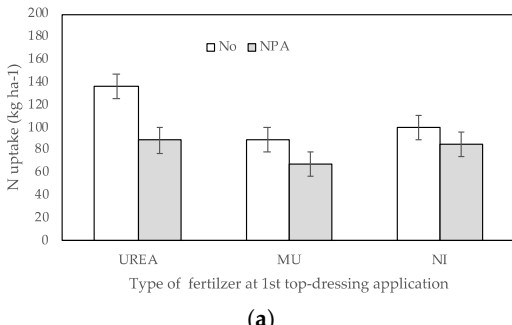 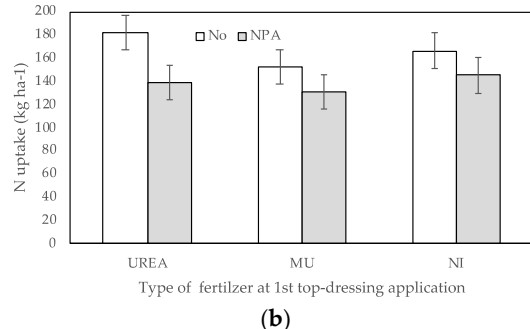

(**a**)                                                               (**b**)

**Figure 4.** N uptake of durum wheat as affected by N source × N rate interaction at the two locations: (**a**) Pisa; (**b**) Arezzo. Vertical bars represent LSD ($p$ = 0.05).

In Pisa, N uptake differed depending on the N rate when methylene urea was applied. With the optimal rate, the crop showed 32% higher N uptake, mainly due to differences in grain N content (69.5 kg ha$^{-1}$ with N$_O$ vs. 50.1 kg ha$^{-1}$ with N$_{Ap}$); interestingly in Pisa MU at the N$_{AP}$ rate, showed the lowest N uptake of all the treatments (Figure 4a).

Finally, the N fertilizer with NI did not show differences between the two N rates at either site.

## 4. Discussion

Our results confirm that seasonal variations in climate influence durum wheat yield [4]. At both NVZs, in the first season, grain yield was 38% and 28% higher than in the second, due to the higher number of spikes produced per unit area. In 2011, the erratic rainfall and lower temperatures of December and January likely impeded the tillers' development, resulting in fewer spikes. Additionally, excessive rainfall during seed filling lowered the mean kernel weight and the N content of grains which thus triggered a low total N uptake, in line with our previous findings in durum wheat [20].

The main purpose of this study was to define N management practices to better synchronize N supply with crop N uptake in durum wheat cropping systems in two nitrate vulnerable zones in central Italy.

Regarding the N source, urea led to 28% higher grain yields than the two slow-release fertilizers at Pisa, while at Arezzo, different N sources led to very similar yields. Therefore, our results did not show any agronomic benefit from using methylene urea or nitrification inhibitors over conventional urea applied at the 1st topdressing event at both sites.

Due to increased rainfall, the high water contents in the soil may have reduced the efficiency of DMPP in the present experiment [21]. Similar previous findings have revealed that yield components and nitrogen use efficiency were not improved by NI in durum wheat [5]. In addition, the increase in NUE for wheat at a range of 9% after the introduction of nitrification and urease inhibitors was not necessarily linked with an increase in grain yield [22].

The results for Pisa highlighted that urea increased the number of spikes per unit area. Using urea at topdressing likely accounted for better N availability in the soil during the spike initiation period, which in durum wheat takes place from the development of the 4th leaf to the stem elongation, probably because of the short period for hydrolyzation of the slow-release N fertilizers, from their application to the spike initiation. Whatever the mechanism is involved, the main feature of the two EEFs fertilizers is that N takes longer to become available to the plants. Under the present experimental conditions, this may have lowered the N available in the soil for durum wheat at the critical stage when the crop N demand increases sharply (just prior to the onset of the most rapid phase of crop growth, i.e., stem elongation). Shortage of N during this period reduced subsequent shoot development and tillering, leading to fewer spikes and, thus, to a lower final grain yield.

This effect was also highlighted by the interaction between the type of fertilizer and the application timing: the two slow-release fertilizers performed better when applied earlier

(at 1st tiller BBCH21), conversely urea produced better results with the later distribution (at 1st node detectable BCCH31).

N uptake was stimulated by urea application in line with increased yields, indicating a positive effect on N use efficiency and supporting our hypothesis mentioned above that MU and MI had released less N than urea. Similar results for durum wheat have also been reported by [23].

There were no evident differences among fertilizers in Arezzo, probably because the lower temperatures recorded may have constrained urea hydrolysis on one side and DMPP action on the other [21,24]. This was confirmed by the similar performances of the different N fertilizers between the two growth stages of 1st topdressing application.

The differences in soil characteristics at the two sites may have been responsible for the different yield responses to N sources. The higher clay content and lower pH at Arezzo may have reduced the effect of the NI and improved MU microbial decomposition, as suggested by Reference [24]. Consequently, the N available in soils from the EEF was similar to that from urea, and the crop obtained similar yields and N uptakes.

Finally, our results highlighted that the two enhanced-efficiency fertilizers showed comparable results. Despite their different mode of action [25], neither of them enabled an N release as fast as urea. Whereas common urea fertilizer likely underwent rapid hydrolysis [26], the conversion of MU to plant-available N is a multistep and longer process involving dissolution and decomposition [24]. For the other fertilizer, the addition of NI slowed down the hydrolysis of urea and, thus, also retarded the nitrification of ammonium [27].

The lower N rates defined by the action programme of the two NVZs had a detrimental effect in both growing seasons and at both sites. Generally, durum wheat with $N_{AP}$ showed reduced biomass and grain yield, and the N content was comparable to $N_O$. However, since reductions in these parameters were proportional, mean HI and NHI were similar.

In the present research, NHI ranged between 68 to 74% and was influenced only by the N rate in Pisa, confirming that it is principally determined by genotype in durum wheat [28–30]. In Pisa, grain yield was improved with the optimal rate only when the topdressing fertilization was performed with urea. Given that when MU or NI was used, the two N rates were comparable, reducing the amount on N given is possible without yield constraints if the N source is selected correctly.

In Arezzo, differences in grain yield due to the N rate were higher when the 1st topdressing fertilization was carried out earlier, probably because the less rainfall at this site could have lowered N-leaching. Consequently, more N remained in the soil for a longer time, and the crop was able to absorb additional N, as confirmed by the greater N uptake.

The optimization of N fertilization is a central issue in the global challenge of meeting increased food demand and protecting the environment in sustainable agriculture. This could be achieved with the 4R approach—right source, right amount, right time, and right placement [31].

We aimed to highlight the effects of N source, timing and rate on durum wheat yields and N uptake in the present research. We concluded that the use of methylene urea and nitrification inhibitors is a potentially attractive approach to improve fertilizer performance. However, without a notable increase in yield and N use efficiency compared to conventional urea, it may not be economically feasible in durum wheat unless there are positive environmental factors such as less leaching of N.

In terms of N management, applying an optimal rate of N is thus more critical to yield and the expected NUE than the time of application. Optimal N fertilization strategies seem to depend on site-specific environmental conditions; in fact, rainfall patterns influenced both N availability and crop uptake.

Our results further indicate that the implementation of the Action Programmes may not be equally effective in the two NVZs, confirming that mitigation measures should be tailored specifically to each NVZ.

**Author Contributions:** Conceptualization, methodology, investigation, S.P. and M.M.; formal analysis, validation, writing—original draft preparation, writing—review and editing, S.P.; funding acquisition, M.M. Both authors have read and agreed to the published version of the manuscript.

**Funding:** This research was partly funded by ARSIA regional agency for development and innovation in agroforestry.

**Institutional Review Board Statement:** Not applicable.

**Informed Consent Statement:** Not applicable.

**Data Availability Statement:** Not applicable.

**Conflicts of Interest:** The authors declare no conflict of interest.

## Abbreviations

EEF, enhanced-efficiency fertilizers; MU, methylene urea; N, nitrogen; NI, nitrification inhibitor; DMPP, 3,4-dimethylpyrazole phosphate; NUE, Nitrogen Use Efficiency; NHI, Nitrogen Harvest Index; NVZ, Nitrate Vulnerable Zone; SRF, slow-release fertilizers.

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
