# Peer review of "Durum Wheat Yield and N Uptake as Affected by N Source, Timing, and Rate in Two Mediterranean Environmentsâ€"

_agronomy, doi:10.3390/agronomy11071299_

Round 1
Reviewer 1 Report
The manuscript “Durum wheat yield and N uptake as affected by N rate, timing 2 and source in two Mediterranean environments” reports a study on the effect of nitrogen rate, time, and source on durum wheat yield and nitrogen uptake. The research was conducted in two nitrate vulnerable zones in Italy covering a period of two growing seasons. A split-split-plot design was adopted in the two experimental fields with nitrogen source as the main treatment, with timing and rate as sub treatments. The introduction presents some important aspects related to nitrogen fertilization, its possible negative impact on the environment, and strategies that can be adopted to reduce these negative impacts leading to the final part of the introduction where the hypothesis and the goals of the research are presented. The paragraph on Materials and Methods presents the study sites and their main characteristics, the experimental fields, and the experimental design describing with sufficient details the treatments applied (type of fertilizer, time of fertilizer application, nitrogen rate). Also, the sampling activity, the data obtained from the samples, and how the data were analyzed are described. In the Results section, information on different aspects that emerged during the research is presented. The section is well organized and covers the main findings of the study while in the final section a discussion based on the findings of the research is presented.
GENERAL COMMENTS AND QUESTIONS
Overall, the manuscript is well written but it could benefit from a review by a native English speaker or by the authors themselves. Also, the manuscript can be improved by paying more attention to details. For instance, around lines 118-121 the intervals of the sizes of soil particles are written without spaces between symbols and numbers, then, on lines 124 and 125, the soil particle sizes are written in different ways.
Line 56-57. The authors listed rate, splitting, timing, and source of N application as factors to consider to improve the NUE and reduce N losses. I think “splitting” can be removed since it is included in rate and timing or the sentence can be reworded.
Line 63-66. This sentence is a little confusing. The authors stated that the split application of N improves NUE but the response of the crop is affected by other factors including the splitting application. Please, revise it.
Line 118-127. When describing the characteristics of the two experimental fields, to what depth are organic matter, total N, and available P and K referred to?
Line 163-166. Why did the authors decide to use different types of fertilizers for the different applications over time instead of using the same type at least for the first two applications (at sowing and first top-dressing)? This question is reinforced by what the authors stated from line 322 to line 328. What would be the effect of using a different type of fertilizer (MU or NI/DMPP) at sowing other than at the first top dressing application?
Have the authors considered monitoring the amount of soil nitrogen available to the crop during the growing season? I think this can be interesting for further developments and it can provide information on how the different types of fertilizers make nitrogen available during the season.
SPECIFIC QUESTIONS AND COMMENTS
Line 13: space missing between the period and the following “To study…”.
Line 15: space missing after the period and the following “ “The trial compared…”.
Line 19: space missing after the period and the following “ Results showed…”.
Line 23: space missing after the period and the following “Different…”.
Line 36-37: European Unit?
Line 44-45: I would replace N utilization efficiency with N use efficiency. The same is valid for other parts of the manuscript.
Line 74-75: NUE was already defined on lines 44-45. No need to define it again here.
Line 84: please check the sentence “enzyme catalyzing the first and rate limiting step of nitrification”.
Line 93: the definition of NHI is missing.
Line 113: Arezzo is repeated two times, in Arezzo hereafter and again in parenthesis after the coordinates.
Line 113-114: I think there is not a “hot Mediterranean” climate (please, correct me if I am wrong and disregards this comment). I would change the sentence to “Mediterranean with hot and dry summers” or I would remove the word “hot”.
Line 118-119: for the silt size, check the unit (in the manuscript m is used instead of mm) and the clay size is missing a zero (in the manuscript 0.02 instead of 0.002).
Line 153: I think the experimental design has 12 treatments (not 36) with three replicates. The authors can use 36 plots instead of treatments.
Line 159: the symbol for the Optimal amount here is different from what was used before (I think here the “o” is subscript while it was not the two times before).
Line 177: the definition of NHI here can be removed after the same will be added on line 93.
Line 218: not sure if it is only a problem in the PDF I have received but, please check the position of the letter “d” to indicate the fourth panel of Figure 1.
Line 223: I suggest making Table 3 a little wider to make it easier to read.
Line 321: I am not convinced that the use of the word “mineralization” is correct in this sentence since the authors are talking about mineral fertilizers while mineralization is usually used for processes that involve soil organic matter.
Author Response
Dear Reviewer,
We are very grateful for your valuable comments - the changes you suggested have undoubtedly improved our manuscript.
We have addressed all the concerns you raised, providing a point-by-point answer on how we handled each suggestion. Our reply (R) is in italics right after each comment (C) in the attached file. Changes made to the text have been highlighted.
Sincerely,

Reviewer 2 Report
This manuscript is aimed to assess different N fertilization strategies in durum wheat cultivation in two Mediterranean environments. Overall, the paper is interesting and the methodology is sound. Nonetheless, some changes are advisable. Publication is recommended, provided that some minors revisions can be made and the English language is improved.
In detail:
- Abstract: I suggest to describe the experiment (“The trial compared…”) by listing the factors according to the adopted experimental design (N source: main plot = main factor; timing: sub-plot; N rate: sub-sub-plot)
- Abstract: in my opinion, the three N sources are urea, MU and urea + DMPP. The third evaluated option is not clear, nitrification inhibitor is mentioned without any reference to the actual N fertilizer.
- Introduction: the sentence at L42-L45 requires supporting literature / citations; the unique characteristics of the Mediterranean climate require clarification (Why are they unique? How are they relevant to durum wheat cultivation and N fertilization?)
- Introduction: it would be ideal to present the factors in the same order of importance. This is not just for a matter of consistency and clarity: the N fertilization rates considered in the study are not that different (especially in Pisa) and fertilization at sowing time is the same in both the considered rates, while the authors say the N use efficiency is particularly critical in autumn. I found this a bit inconsistent with the aim of the study, that is focused on side dressing. I suggest to introduce N rate as a factor related to side dressing and to explain it later in the Introduction (as a sub-sub-factor, as it is in the experimental design)
- The sentence at L61-62 does not well clarify what the “optimal rates” for durum wheat fertilization are. I must say that the difference between the two N rates should be better clarified throughout the whole paper. A reference to the formula used to calculate N rate in NVZ areas will for sure help.
- L93: the abbreviation “NHI” is not explained.
- L95: sentence about protein content is unclear to me
- Materials and methods: I suggest to include the USDA classification of soil texture of the two soils.
- L112-113: “Pisa henceforward”, “Arezzo hereafter” sound redundant to me. It’s Pisa and Arezzo, sufficiently clear to me
- L156: Where the yields used to calculate the optimal rates come from? Long-term averages? Crop models? The link with the cited literature is not entirely clear to me, perhaps some more details can be given, in order to allow the reader to quickly understand.
- Results, L201-203: this sentence is unclear, especially regarding the reference to further publications
- L208: the abbreviation “VAP” is not defined in the Methods, I suggest to define it at an earlier stage in the paper
- L266: is this referred to NO instead of NAP?
- L303: Why “so far”?
- BBCH31, why is it so relevant, despite being not so far from BBCH32? I suggest to discuss this.
A more general remark: the manuscript would greatly benefit from some discussion and introduction of the environmental relevance of N use efficiency in durum wheat cultivation and on how the findings of the study could be interpreted in the light of reduced leaching and N losses. The study is well done, but it is advisable to delve a bit more into the following "big questions":
- What is the fertilization strategy the authors can advise? Is there a difference between the two environments? Is this possibly due to different rainfall regimes? Is the N “spoon feeding” always more advisable than nitrification inhibitors?
- Do the authors can expect different effects on N leaching depending on the fertilization strategy adopted, with specific reference to the treatments they investigated?
- Can the authors give some advice on the implementation of the Nitrate Directive? Not only yields are higher in the NO strategy (and this is pretty straightforward), but also N use efficiency is often higher. Can the authors further discuss this? Would they expect any differences if the difference between applied N and recovered N was considered? (unfortunately, the study does not have a “zero nitrogen” treatment, the so called “agronomic efficiency” of N fertilization cannot be calculated; however, the literature is not scarce on this and some final remarks / insights for discussion can be provided
Author Response
Dear Reviewers,
We are very grateful for your valuable comments - the changes you suggested have undoubtedly improved our manuscript.
We have addressed all the concerns you raised, providing a point-by-point answer on how we handled each suggestion. Our reply (R) is in italics right after each comment (C) in the attached file. Changes made to the text have been highlighted.
Sincerely,
